# A Novel Approach for Constrained Optimization in Graphical Models

**Sara Rouhani, Tahrima Rahman and Vibhav Gogate**
The University of Texas at Dallas
{sara.rouhani,tahrima.rahman,vibhav.gogate}@utdallas.edu

## Abstract

We consider the following constrained maximization problem in discrete probabilistic graphical models (PGMs). Given two (possibly identical) PGMs $\mathcal{M}_1$ and $\mathcal{M}_2$ defined over the same set of variables and a real number $q$, find an assignment of values to all variables such that the probability of the assignment is maximized w.r.t. $\mathcal{M}_1$ and is smaller than $q$ w.r.t. $\mathcal{M}_2$. We show that several explanation and robust estimation queries over graphical models are special cases of this problem. We propose a class of approximate algorithms for solving this problem. Our algorithms are based on a graph concept called $k$-separator and heuristic algorithms for multiple choice knapsack and subset-sum problems. Our experiments show that our algorithms are superior to the following approach: encode the problem as a mixed integer linear program (MILP) and solve the latter using a state-of-the-art MILP solver such as SCIP.

## 1 Introduction

This paper is about solving the following combinatorial optimization problem: given a set of discrete random variables **X** and two possibly identical probabilistic graphical models (cf. [7, 20]) or log-linear models defined over **X**, find the most likely assignment to all variables w.r.t. one of the log-linear models such that the weight (or probability) of the assignment is smaller than a real number $q$ w.r.t. the second model. We call this task *constrained most probable explanation (CMPE)* problem. CMPE is NP-hard in the strong sense and thus it cannot have a fully polynomial time approximation scheme (or FPTAS) unless P = NP. However, it is only weakly NP-hard when the log-linear models exhibit certain properties such as their features are conditionally independent of each other (e.g., Naive Bayes, Logistic Regression, etc.) or all connected components in the two models have bounded number of variables (e.g., small-world graphs [33]) or both given a small subset of variables. We exploit this weak NP-hardness property to develop approximation algorithms for CMPE.

We are interested in solving the CMPE problem because several estimation, prediction and explanation tasks can be reduced to CMPE. For example, the nearest assignment problem (NAP) [30]—a problem that is related to the nearest neighbors problem—which requires us to find an assignment of values to all variables such that the probability of the assignment is as close as possible to an input value $q$, can be reduced to CMPE. Similarly, the problem of computing the most likely assignment to all unobserved variables given evidence such that a log-linear model makes a particular (single class or multi-label) classification decision can be reduced to CMPE. This problem is the optimization analog of the same decision probability problem [5, 6, 29]. Other problems that reduce to CMPE include finding diverse $m$-best most probable explanations [1, 12], the order statistics problem [32] and the adversarial most probable explanation problem.

We propose a novel approach that combines graph-based partitioning techniques with approximation algorithms developed in the multiple choice knapsack problem (MCKP) literature [22, 27, 31] for solving CMPE. MCKP is a generalization of the 0/1 knapsack problem in which you are given a knapsack with capacity $q$ and several items which are partitioned into bins such that each item is

associated with two real numbers which denote its profit and cost respectively; the problem is to find a collection of items such that exactly one item from each bin is selected, the total cost of the selected items does not exceed $q$ and the total profit is maximized. We show that when the combined primal graph, which is obtained by taking a union of the edges of the primal graphs of the two graphical models, has multiple connected components and each connected component has a constant number of variables, CMPE is an instance of *bounded* MCKP. We exploit the fact that such bounded MCKPs are weakly NP-hard and can be solved efficiently using approximate algorithms with guarantees [19].

On graphical models in which the number of variables in each connected component is not bounded by a constant, we propose to condition on variables, namely remove variables from the combined primal graph, until the number of variables in each connected component is bounded (above) by a constant $k$. We formalize this conditioning approach using a graph property called $k$-separator [2]. A $k$-separator of a graph $G$ is a subset of vertices which when removed from $G$ yields a graph $G'$ such that the number of vertices in each connected component of $G'$ is at most $k$. Our proposed conditioning (local search) algorithm solves the sub-problem over the $k$-separator via local/systematic search and the sub-problem over the connected components given an assignment to all variables in the $k$-separator using MCKP methods. Our algorithm yields a heuristic approximation scheme with performance guarantees when the size of the $k$-separator is bounded. In practice, it is likely to yield high quality estimates when the size of the $k$-separator is small (e.g., in small-world networks [33]).

We performed a detailed experimental evaluation comparing the impact of increasing $k$ on the quality of estimates computed by our proposed method. As a strong baseline, we encoded CMPE as a mixed integer linear program (MILP) and used a state-of-the-art open source MILP solver called SCIP [13]. We used various benchmark graphical models used in past UAI competitions [11, 16]. Our experiments show that somewhat counter intuitively most CMPE problems are easy in that our method yields close to optimal solutions within seconds even when $k$ is small (we expect our method to be more accurate when $k$ is large). Hard instances of CMPE arise when $q$ is close to the unconstrained maximum or when the parameters of the graphical model are extreme. Such hard instances do benefit from using a large value of $k$ while easy instances do not. Our experiments clearly show that our proposed algorithm is superior to SCIP.

## 2 Preliminaries and Notation

Let $\mathbf{X} = \{X_1, \ldots, X_n\}$ denote a set of discrete random variables and $D_i = \{1, \ldots, d\}$ be the domain of $X_i$, namely we assume that each variable $X_i$ takes values from the set $D_i$. A graphical model or a Markov network denoted by $\mathcal{M}$ is a triple $\langle \mathbf{X}, \mathbf{f}, G \rangle$ where: (1) $\mathbf{f} = \{f_1, \ldots, f_m\}$ is a set of *log-potentials* where each log-potential $f_i$ is defined over a subset $S(f_i) \subseteq \mathbf{X}$ called the scope of $f_i$, and (2) $G(\mathbf{V}, \mathbf{E})$ is an undirected graph called the *primal graph* where $\mathbf{V} = \{V_1, \ldots, V_n\}$ and $\mathbf{E} = \{E_1, \ldots, E_t\}$ denote the set of vertices and edges in $G$ respectively. $G$ has (exactly) one vertex $V_i$ for each variable $X_i$ in $\mathbf{X}$ and an edge $E_j = (V_a, V_b)$ if the corresponding variables $X_a$ and $X_b$ appear in the scope of a function $f$ in $\mathbf{f}$. $\mathcal{M}$ represents the following probability distribution:

$$P_{\mathcal{M}}(\mathbf{x}) = \frac{1}{Z_{\mathcal{M}}} \exp \left( \sum_{f \in \mathbf{f}} f(\mathbf{x}_{S(f)}) \right) \text{ where } Z_{\mathcal{M}} = \sum_{\mathbf{x}} \exp \left( \sum_{f \in \mathbf{f}} f(\mathbf{x}_{S(f)}) \right)$$

where $\mathbf{x} = (x_1, \ldots, x_n)$ is an assignment of values to all variables in $\mathbf{X}$ and $\mathbf{x}_{S(f)}$ denotes the projection of $\mathbf{x}$ on the set $S(f)$. Note that $\mathbf{x} \in D$ where $D = D_1 \times \ldots \times D_n$ is the Cartesian product of the domains. $Z_{\mathcal{M}}$ is normalization constant called the partition function. For brevity, henceforth, we will write $f(\mathbf{x}_{S(f)})$ as $f(\mathbf{x})$.

Given an assignment $\mathbf{x}$ of values to all variables in $\mathbf{X}$ of a graphical model $\mathcal{M} = \langle \mathbf{X}, \mathbf{f}, G \rangle$, we call $\sum_{f \in \mathbf{f}} f(\mathbf{x})$ the *weight* of $\mathbf{x}$ w.r.t. $\mathcal{M}$. We focus on the following constrained optimization problem, which we call the constrained most probable explanation problem (CMPE). Given two graphical models $\mathcal{M}_1 = \langle \mathbf{X}, \mathbf{f}_1, G_1 \rangle$ and $\mathcal{M}_2 = \langle \mathbf{X}, \mathbf{f}_2, G_2 \rangle$ and a real number $q$, find an assignment $\mathbf{X} = \mathbf{x}$ such that the weight of $\mathbf{x}$ w.r.t. $\mathcal{M}_1$ is maximized and is bounded above by $q$ w.r.t. $\mathcal{M}_2$. Formally,

$$\max_{\mathbf{x}} \sum_{f \in \mathbf{f_1}} f(\mathbf{x}) \quad \text{s.t.} \quad \sum_{g \in \mathbf{f_2}} g(\mathbf{x}) \leq q \tag{1}$$

## 2.1 Multiple Choice Knapsack and Subset Sum Problems

Given $n$ items where each item $i$ has an associated cost $c_i$ and profit $p_i$, and a container/knapsack having capacity $q$, the Knapsack problem (KP) is to select a subset of the items such that the total cost does not exceed $q$ and the total profit is maximized. Let the items be denoted by the integers 1, 2,..., $n$, and let $X_i$ be a Boolean variable taking the value 1 if item $i$ is selected and 0 otherwise. Let $\mathbf{x} = (x_1, \ldots, x_n)$ denote a 0/1 assignment to all the Boolean variables. Then, the Knapsack problem can be mathematically stated as: $\max_{\mathbf{x}} \sum_{i=1}^n p_i x_i$ s.t $\sum_{i=1}^n c_i x_i \leq q$. The subset-sum problem (SSP) is a special case of the knapsack problem where profit $p_i$ equals the cost $c_i$ for all items $i$.

The multiple choice Knapsack problem (MCKP) is a generalization of KP in which the items are partitioned into bins and the constraint is that exactly one item from each bin must be chosen. Let $m$ be the number of bins and $N_1, \ldots, N_m$ denote the number of items in each bin. Let $i = 1, \ldots, m$ index the bins and $j = 1, \ldots, N_i$ index the items in bin $i$. Let $c_{ij}$ and $p_{ij}$ denote the cost and profit respectively of the $j$-th item in the $i$-th bin. Let $X_{ij}$ be a Boolean variable taking the value 1 if $j$-th item in the $i$-th bin is selected and 0 otherwise. Let $x_{ij}$ denote the 0/1 assignment to the Boolean variable $X_{ij}$ and $\mathbf{x}$ denote a 0/1 assignment to all the Boolean variables. Then, the MCKP is given by

$$\max_{\mathbf{x}} \sum_{i=1}^m \sum_{j \in N_i} p_{ij} x_{ij} \quad \text{s.t.} \quad \sum_{i=1}^m \sum_{j \in N_i} c_{ij} x_{ij} \leq q \quad \text{and} \quad \sum_{j \in N_j} x_{ij} = 1, \quad i = 1, \ldots, m \qquad (2)$$

The multiple choice subset-sum problem (MCSSP) is a special case of MCKP where $p_{ij}$ equals $c_{ij}$ for all $i, j$. We focus on a bounded version of MCKP where $N_i$ is bounded by a constant for all $i$.

All of the aforementioned problems, KP, SSP, MCKP and MCSSP are NP-hard. However, they can be solved in pseudo-polynomial time via a dynamic programming algorithm if the profits and weights are integers. There exists a vast literature on algorithms for solving these problems with specific interest from the operations research community. The different types of algorithms presented in literature include branch and bound algorithms [9, 10, 26, 31], local search algorithms, dynamic programming algorithms [9, 18, 27], heuristic algorithms with performance guarantees [10, 14] and fully polynomial time approximate schemes (FPTAS) [4, 22]. The purpose of this paper is to show that these algorithms can be leveraged, in addition to graph-based methods to solve the CMPE task.

## 3 Applications of CMPE

In this section, we show that the nearest assignment problem and the decision preserving most probable assignment problem can be reduced to CMPE.

### 3.1 The Nearest Assignment Problem

Given a graphical model $\mathcal{M} = \langle \mathbf{X}, \mathbf{f}, G \rangle$ and a real number $q$, the nearest assignment problem (NAP) is to find an assignment $\mathbf{x}$ to all variables in $\mathbf{X}$ such that $|q - \sum_{f \in \mathbf{f}} f(\mathbf{x})|$ is minimized. We can express NAP as CMPE using the following transformation. For each function $f \in \mathbf{f}$, let $g$ be a function defined as follows: $g(\mathbf{y}) = f(\mathbf{y}) - q/m$, where $\mathbf{y}$ is an assignment of values to all variables $\mathbf{Y} = S(f)$ and $m$ is the number of log-potentials, namely $m = |\mathbf{f}|$. Let $\mathbf{x}^l$ and $\mathbf{x}^u$ be two assignments defined as follows:

$$\mathbf{x}^l = \arg\max_{\mathbf{x}} \sum_{i=1}^m g_i(\mathbf{x}) \text{ s.t. } \sum_{i=1}^m g_i(\mathbf{x}) \leq 0 \quad \text{and} \quad \mathbf{x}^u = \arg\max_{\mathbf{x}} \sum_{i=1}^m -g_i(\mathbf{x}) \text{ s.t. } \sum_{i=1}^m -g_i(\mathbf{x}) \leq 0$$

We can show that:

**Proposition 1.** $\arg\min_{\mathbf{x}} |q - \sum_{f \in \mathbf{f}} f(\mathbf{x})|$ *where* $\mathbf{x} \in \{\mathbf{x}^l, \mathbf{x}^u\}$ *is the nearest assignment.*

By inspection, the expressions for $\mathbf{x}^l$ and $\mathbf{x}^u$ are CMPE tasks. Thus NAP can be solved by solving two CMPE tasks. Rouhani et al. [30] describe an approximation algorithm that uses a 0-cutset [3] to solve NAP. 0-cutsets are equivalent to 1-separators. Thus, our general-purpose algorithm can be seen as a generalization of Rouhani et al.'s approach. Moreover, their approach is not applicable to variables having non-binary domains while our proposed approach does not have such limitations. Finally, Rouhani et al.'s approach can be used if and only if the graphical model in the objective function

is identical to the graphical model in the cost constraint. Our approach allows different graphical models to be present in the objective and cost constraint. It turns out that NAP instances are one of the hardest CMPE problems; since they have subset-sum type constraints.

### 3.2 The Decision Preserving Most Probable Assignment

Consider the following problem from robust estimation or decision theory that is useful in interactive settings for solving human-machine tasks. You are given a log-linear model with a few observed variables $\mathbf{E}$ and a decision variable $C$ (or a small subset $\mathbf{C} \subseteq \mathbf{X}$) that takes values from the domain $\{0, 1\}$. Suppose that you have made the decision $C = c$ given evidence $\mathbf{E} = \mathbf{e}$ because the weight of the partial assignment $(c, \mathbf{e})$ is higher than that of $(1 - c, \mathbf{e})$. Your task is to find the most probable assignment to all unobserved variables $\mathbf{H}$ such that the same decision will be made.[1] Formally, given a graphical model $\mathcal{M} = \langle \mathbf{X}, \mathbf{f}, G \rangle$ where $\mathbf{X} = \{C\} \cup \mathbf{E} \cup \mathbf{H}$, we have to solve

$$\max_{\mathbf{h}} \sum_{f \in \mathbf{f}} f(\mathbf{h}, c, \mathbf{e}) \quad \text{s.t.} \quad \sum_{f \in \mathbf{f}} f(\mathbf{h}, c, \mathbf{e}) \geq \sum_{f \in \mathbf{f}} f(\mathbf{h}, 1 - c, \mathbf{e}) \tag{3}$$

Let $\mathbf{g} = \{f(\mathbf{h}, 1 - c, \mathbf{e}) - f(\mathbf{h}, c, \mathbf{e}) | f \in \mathbf{f}\}$. Then, we can rewrite Eq. (3) as:

$$\max_{\mathbf{h}} \sum_{f \in \mathbf{f}} f(\mathbf{h}, c, \mathbf{e}) \quad s.t. \quad \sum_{g \in \mathbf{g}} g(\mathbf{h}, \mathbf{e}) \leq 0 \tag{4}$$

By inspection, it is clear that Eq. (4) is an instance of the CMPE problem.

The generated assignment $\mathbf{h}$ can then be sent to an expert (human in the human-machine task) for analyzing the robustness of the decision. As mentioned earlier, the problem just described is an optimization analog of the same decision probability problem [5, 6] where one seeks to find the probability that the same decision will be made after observing all the unobserved variables.

## 4 Our Approach

### 4.1 CMPE with Multiple Connected Components

We show that if the combined primal graph associated with $\mathcal{M}_1$ and $\mathcal{M}_2$ has multiple connected components and the number of variables in each connected component is bounded by a constant $k$, then CMPE can be encoded as a bounded MCKP. We begin by defining a combined primal graph.

**Definition 4.1.** A *combined primal graph* of two graphical models $\mathcal{M}_1 = \langle \mathbf{X}, \mathbf{f}_1, G_1 \rangle$ and $\mathcal{M}_2 = \langle \mathbf{X}, \mathbf{f}_2, G_2 \rangle$ is a graph $G(\mathbf{V}, \mathbf{E})$ such that $G$ has a vertex $V_k$ for each variable $X_k$ in $\mathbf{X}$ and an edge $E_t = (V_a, V_b)$ if the corresponding variables $X_a$ and $X_b$ appear in the scope of $f_i \in \mathbf{f}_1$ or $f_j \in \mathbf{f}_2$.

Let $G$ denote the combined primal graph of $\mathcal{M}_1$ and $\mathcal{M}_2$. Let $c$ denote the number of connected components of $G$. Let $\mathbf{X}_i$ denote the set of variables (corresponding to the set of vertices) in the $i$-th connected component of $G$ ($1 \leq i \leq c$). Let $g_1, \ldots, g_c$ and $h_1, \ldots, h_c$ denote the functions obtained from $\mathcal{M}_1$ and $\mathcal{M}_2$ s.t. for $i = 1, \ldots, c$

$$g_i(\mathbf{x}_i) = \sum_{f \in \mathbf{f}_1 : S(f) \subseteq \mathbf{X}_i} f(\mathbf{x}_i) \quad \text{and} \quad h_i(\mathbf{x}_i) = \sum_{f \in \mathbf{f}_2 : S(f) \subseteq \mathbf{X}_i} f(\mathbf{x}_i)$$

**Encoding 4.2.** Given a collection of functions $\mathbf{g} = \{g_1, \ldots, g_c\}$, $\mathbf{h} = \{h_1, \ldots, h_c\}$ such that no two functions in $\mathbf{g}$ (and $\mathbf{h}$) share any variables and $S(g_i) = S(h_i)$ for $1 \leq i \leq c$, and a real number $q$, we can construct a MCKP, denoted by $P$ as follows. We start with an empty MCKP. Then we create an item for each entry $j$ in each function $g_i$ (or $h_i$). For each component indexed by $i$, we add a bin (indexed by $i$) to $P$ (thus there are $c$ bins) and add all items corresponding to the entries in function $g_i$ (or $h_i$) to the $i$-th bin. We attach a knapsack with capacity $q$ to $P$. The profit and cost of each item $(i, j)$ in $P$ equals the value of corresponding $j$-th entry in the functions $g_i$ and $h_i$ respectively.

Fig. 1 illustrates the process of converting a given CMPE problem to MCKP using Encoding 4.2. It is easy to show that Encoding 4.2 is correct, namely we can construct a (feasible or optimal) solution to the CMPE problem from a solution of the corresponding MCKP. Formally,

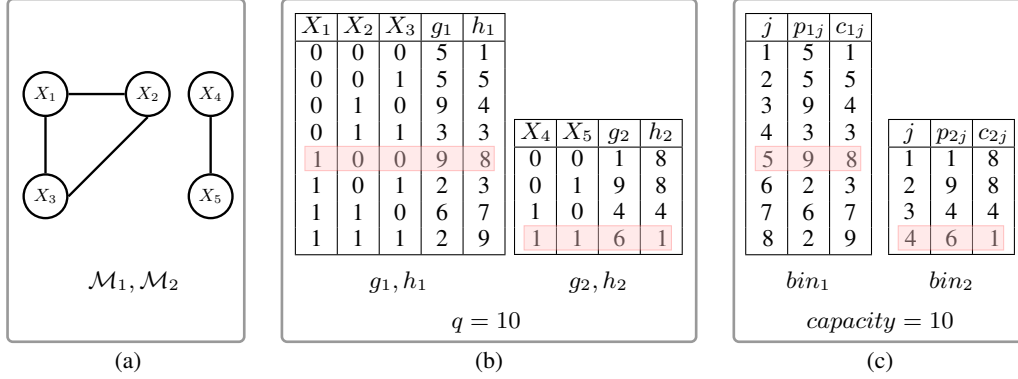

Figure 1: (a) Combined primal graph of two graphical models $\mathcal{M}_1, \mathcal{M}_2$ having 5 binary variables $\{X_1, \ldots, X_5\}$. The graph has two connected components $\{X_1, X_2, X_3\}$ and $\{X_4, X_5\}$. (b) CMPE problem over $\mathcal{M}_1$ and $\mathcal{M}_2$ with $q = 10$, example log-potentials $g_1, g_2$ computed from $\mathcal{M}_1$, and example log-potentials $h_1, h_2$ computed from $\mathcal{M}_2$. Values of the two potentials are generated randomly. (c) Multiple choice knapsack problem (MCKP) encoding of the CMPE problem given in (b) (see Encoding 3.2). The MCKP has 2 bins; the first bin has 8 items while the second has 4 items with $capacity = q = 10$. Optimal solution to the MCKP and the corresponding optimal solution to the CMPE problem is highlighted in red.

**Proposition 2.** *(Equivalence) Let $P$ be a MCKP constructed from a CMPE problem, denoted by $R$ using Encoding 4.2. Then there exists a one-to-one mapping between every feasible (or optimal) solution of $P$ and $R$. Moreover, a feasible (or optimal) solution to $R$ can be constructed from a feasible (or optimal) solution to $P$ in time that scales linearly with the size of $\mathcal{M}_1$ and $\mathcal{M}_2$.*

Since the number of items in each bin $i$ equals the number of entries in $g_i$, the number of items in bin $i$ is exponential in the number of variables in the scope of $g_i$, namely it equals $\exp(|\mathbf{X}_i|)$. Thus Encoding 4.2 will yield a bounded MCKP if $|\mathbf{X}_i|$ is bounded by a constant for all $i$.

## 4.2 A Conditioning Algorithm Based on k-separators

Graphical models typically encountered in practice will have just one connected component and therefore the approach presented in the previous subsection will be exponential in $n$ (number of variables). To address this issue, we propose to condition on variables until the size of each bin in the encoded MCKP is bounded by a constant. We formalize this approach using the following definition:

**Definition 4.2.** ($k$-separators) Given a graph $G(\mathbf{V}, \mathbf{E})$ and an integer $k \geq 1$, a $k$-separator of a graph is a set of vertices $\mathbf{S} \subset \mathbf{V}$ such that each connected component of a graph $G'$ obtained from $G$ by removing $\mathbf{S}$ has at most $k$ vertices. A $k$-separator $\mathbf{S}$ is *minimal* when no proper subset of $\mathbf{S}$ is a $k$-separator. A $k$-separator $\mathbf{S}$ is *optimal* if there does not exist a $k$-separator $\mathbf{S}'$ such that $|\mathbf{S}'| < |\mathbf{S}|$.

In practice, we want $k$-separators that are optimal. Unfortunately, finding optimal $k$-separators is a NP-hard problem [2] and therefore we will use greedy algorithms that yield minimal $k$-separators.[2] One such greedy algorithm is to iteratively remove a vertex having the maximum degree from each connected component having more than $k$ vertices until all components have at most $k$ vertices.

Given a $k$-separator $\mathbf{S} \subseteq \mathbf{X}$ obtained from the combined primal graph of $\mathcal{M}_1$ and $\mathcal{M}_2$, we can use the following conditioning algorithm to yield a solution to CMPE. For each assignment of values $\mathbf{s}$ to $\mathbf{S}$, we get a CMPE sub-problem $R_\mathbf{s}$ such that the MCKP encoding of $R_\mathbf{s}$, denoted by $P_\mathbf{s}$ is bounded. Specifically, $P_\mathbf{s}$ is such that the number of items in each bin is bounded by $\exp(k)$ while the number of bins equals the number of components $c$ which in turn is bounded above by $(|\mathbf{X}| - |\mathbf{S}|)$. We can either explore the space of assignments to $\mathbf{S}$ systematically using branch and bound search or via simulation techniques such as random sampling and local search. Both approaches will yield anytime algorithms whose performance improves with time. As before, we can solve each MCKP sub-problem using advanced MCKP algorithms presented in literature on knapsack problems (cf. [19]).

The above setup and discussion yields Algorithm 1, which is an anytime algorithm for approximately solving the CMPE problem. The algorithm begins by heuristically selecting a minimal $k$-separator $\mathbf{S}$

**Algorithm 1** ANYTIME-CMPE $(\mathcal{M}_1, \mathcal{M}_2, q, k)$

---

**Input:** Two Markov networks $\mathcal{M}_1 = \langle \mathbf{X}, \mathbf{f}_1, G_1 \rangle$ $\mathcal{M}_2 = \langle \mathbf{X}, \mathbf{f}_2, G_2 \rangle$, a real number $q$ and an integer $k$
**Output:** An estimate of the CMPE problem defined over $(\mathcal{M}_1, \mathcal{M}_2, q, k)$.
**Begin:**
1: Heuristically select a minimal $k$-separator $\mathbf{S} \subset \mathbf{X}$ using the combined primal graph $G$
2: $G' = $ graph obtained by removing $\mathbf{S}$ from $G$. Let $c$ denote the number of connected components of $G'$ and let $\mathbf{X}_i$ denote the set of variables in the $i$-th connected component of $G'$
3: $best = -\infty$
4: **repeat**
5:     Generate an assignment $\mathbf{s}$ of $\mathbf{S}$ via random sampling or local search or systematic enumeration
6:     $q_\mathbf{s} = q - \sum_{f \in \mathbf{f}_2 : S(f) \subseteq \mathbf{S}} f(\mathbf{s})$
7:     **for** $i = 1$ to $c$ **do**
8:         Compute $g_i(\mathbf{x}_i) = \sum_{f \in \mathbf{f}_1 : S(f) \subseteq \mathbf{X}_i} f(\mathbf{x}_i, \mathbf{s})$
9:         Compute $h_i(\mathbf{x}_i) = \sum_{f \in \mathbf{f}_2 : S(f) \subseteq \mathbf{X}_i} f(\mathbf{x}_i, \mathbf{s})$
10:     Construct a MCKP $P_\mathbf{s}$ from $\{g_1, \ldots, g_c\}$, $\{h_1, \ldots, h_c\}$ and $q_\mathbf{s}$ using Encoding 4.2
11:     Use the Greedy MCKP method of [14] to solve $P_\mathbf{s}$ and store the objective function value in $current_g$
12:     $current = current_g + \sum_{f \in \mathbf{f}_1 : S(f) \subseteq \mathbf{S}} f(\mathbf{s})$
13:     **if** $P_\mathbf{s}$ is feasible and $current > best$ **then** $best = current$
14: **until** there is time
15: **return** $best$

**End.**

---

of the combined primal graph $G$ (line 1). Then it searches, either via random sampling or local search or systematic enumeration, over the assignments $\mathbf{s}$ of $\mathbf{S}$ (lines 4–16). To perform local search, it selects a neighbor of the current state having the highest value of the objective function or makes a random move if the algorithm is stuck in local maxima. A neighbor of an assignment $\mathbf{s}$ is an assignment $\mathbf{s}$' that differs from $\mathbf{s}$ in assignment to only one variable. Then, in lines 6-10, it converts the CMPE sub-problem obtained after conditioning on the assignment $\mathbf{S} = \mathbf{s}$ to MCKP, as detailed in Encoding 4.2. The CMPE sub-problem is constructed by updating $q$ appropriately (line 6) and computing the functions $g_i$ and $h_i$ for each component $i$ of $G'$ (the graph obtained by removing $\mathbf{S}$ from the combined primal graph $G$) (lines 7-9). The algorithm solves the MCKP using a greedy approach (see [14, 19] for details) (line 11) and updates the best solution computed so far if the current solution has a higher value for the objective function. The algorithm stops when a user specified time bound is reached and returns the best value of the objective function found so far (line 16).

### 4.3 Computational Complexity of ANYTIME-CMPE

Since the size of each function $g_i$ and $h_i$ is bounded exponentially by $k$, the time complexity of lines 7-10 of Algorithm ANYTIME-CMPE is $O(c \exp(k))$. Since the time complexity of the greedy MCKP method [14] is linear in the number of items, and the number of items is bounded by $O(c \exp(k))$, the time complexity of line 11 is also bounded by $O(c \exp(k))$. Thus, the overall time complexity of lines 4-14 is $O(n + c \exp(k))$. If a systematic enumeration method is used for generating the assignment of values to $\mathbf{S}$ then the worst case time complexity of ANYTIME-CMPE is $O((n + c \exp(k)) \times \exp(|\mathbf{S}|))$.

Since the greedy algorithm of Gens and Levner [14] has a performance factor of $4/5$, ANYTIME-CMPE with systematic enumeration yields a polynomial time approximation scheme with a performance factor of $4/5$ when $k$ and $|\mathbf{S}|$ are bounded by a constant (e.g., in some small-world graphs [33]). Algorithm ANYTIME-CMPE can also be used to yield a fully polynomial time approximation scheme (FPTAS) by using a FPTAS algorithm for MCKP [15, 21] in lieu of the greedy algorithm in line 11 (when $k$ and $|\mathbf{S}|$ are bounded by a constant). Note that these guarantees are the best we can hope for because CMPE is strongly NP-hard and is unlikely to have an FPTAS algorithm unless P=NP.

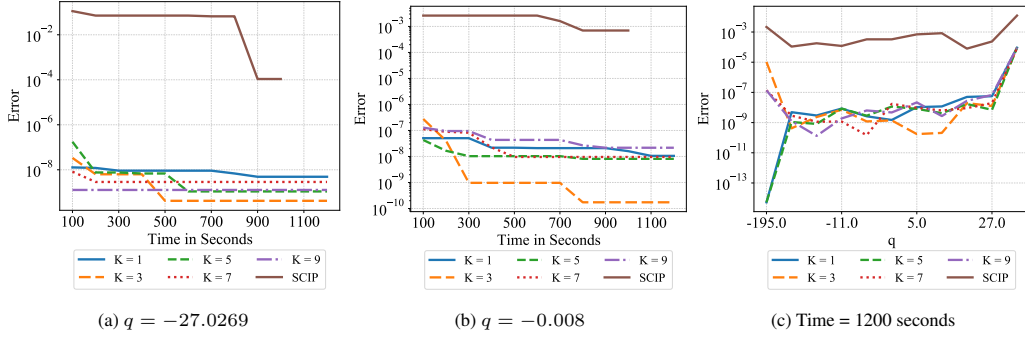

(a) $q = -27.0269$      (b) $q = -0.008$      (c) Time = 1200 seconds

Figure 2: Easy problems: Results on DBN_16 Markov network having 44 variables and 528 potentials.

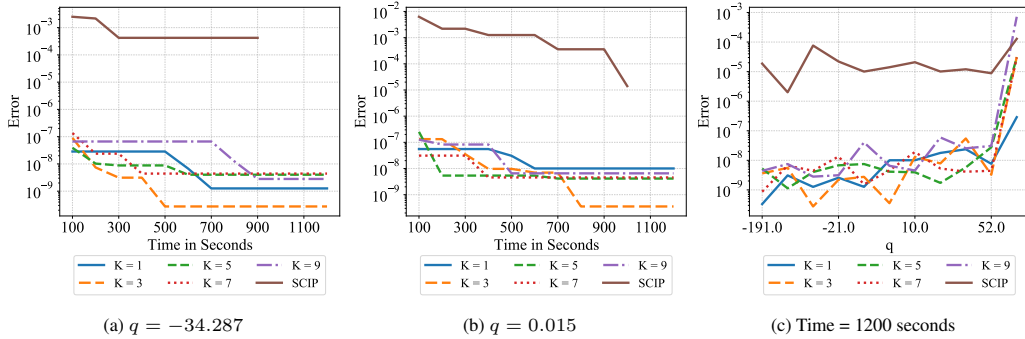

(a) $q = -34.287$      (b) $q = 0.015$      (c) Time = 1200 seconds

Figure 3: Easy problems: Results on Grids_11 Markov network having 100 variables and 300 potentials.

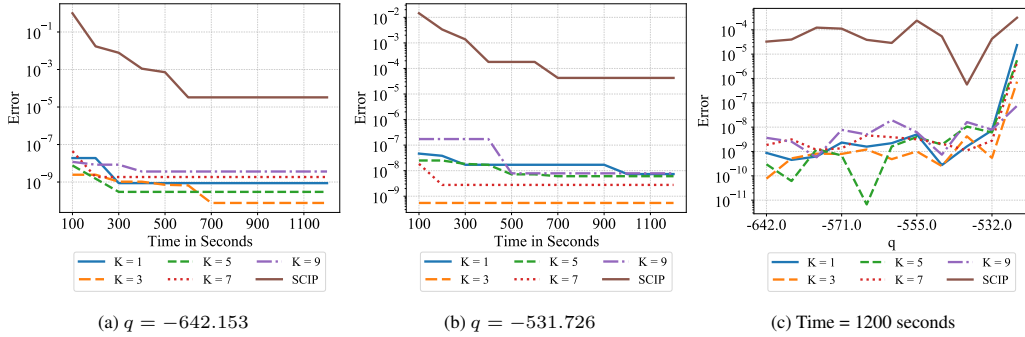

(a) $q = -642.153$      (b) $q = -531.726$      (c) Time = 1200 seconds

Figure 4: Easy problems: Results on Segmentation_12 Markov network with 229 variables and 851 potentials.

## 5 Experiments

### 5.1 Setup

We compared the performance of Algorithm ANYTIME-CMPE with SCIP [13], a state-of-the-art open source mixed integer linear programming (MILP) solver.[3] We evaluated the impact of increasing $k$ and time on the performance of ANYTIME-CMPE. We experimented with the following five values of $k$: $\{1, 3, 5, 7, 9\}$. For each $k$, we ran our algorithm on each probabilistic network for 1200 seconds. SCIP was also run for 1200 seconds on each network.

**Implementation Details.** We used restart-based local search to perform search over the value assignments to **S**. Specifically, our implementation makes locally optimal moves if it improves the evaluation score. Otherwise, the local search is stuck in local maxima and we make a random move. We implemented the greedy MCKP algorithm of [14] to solve $P_{\mathbf{s}}$. We improve the greedy solution further by performing local search over $P_{\mathbf{s}}$ until a local maxima is reached. We used the max-degree heuristic outlined in section 4.2 to select a minimal $k$-separator.

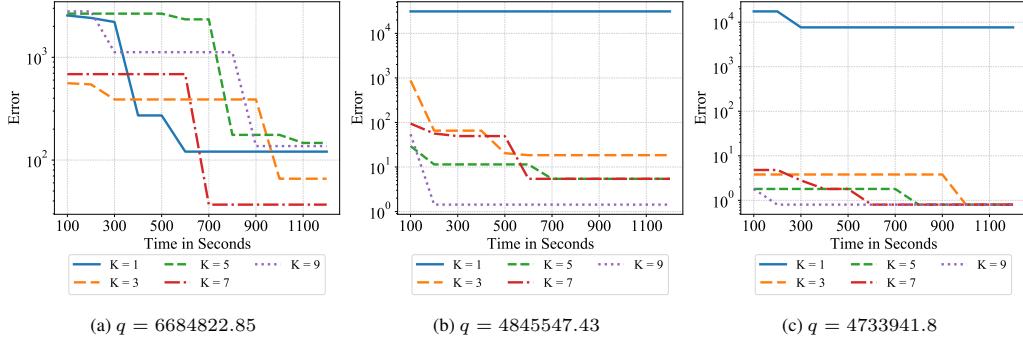

(a) $q = 6684822.85$        (b) $q = 4845547.43$        (c) $q = 4733941.8$

Figure 5: Hard problems: Results on (a) Grids_17 Markov network with 400 variables and 1160 potentials, (b) Segmentation_12 Markov network with 229 variables and 851 potentials, and (c) Segmentation_14 Markov network with 226 variables and 845 potentials.

**Benchmarks and Methodology.** We experimented with the following benchmark graphical models, available from the UAI 2010 and 2014 competitions [11, 16]: (1) Large Dynamic Bayesian Networks, (2) Ising models and (3) Image Segmentation networks. For each benchmark network, we selected ten $q$ values as follows. We generated a million assignments uniformly at random and divided their weights into 10 quantiles (deciles). Then, we selected a random weight from each of the 10 quantiles as a value for $q$. We found that most CMPE problems generated this way were easy problems in that the maximum value of the objective function (or close to it) was reached quickly by all of our local search algorithms. Similar observations have been made in the literature on knapsack problems [28]. Therefore, in order to generate hard CMPE problems, we made the following modifications: (1) for each network, we kept the network structure the same but changed the parameters by sampling each parameter from the range $[0, 10000]$; (2) we selected values of $q$ that are close to the weight of the (unconstrained) most probable assignment; and (3) we focused on multiple choice subset sum problems, namely we chose $\mathcal{M}_1 = \mathcal{M}_2$. We use the quantity $q - o$, which we call error to measure performance where $o$ is value of the objective function output by the competing algorithms. Note that the maximum value of the objective function is bounded by $q$ (since we are solving hard subset sum type problems) and therefore smaller the error, better the algorithm.

## 5.2 Results

We present results for two classes of problems: (1) relatively *easy* subset-sum type problems on the original networks; and (2) *hard* subset-sum type problems on the modified networks (only the parameters are modified as described above) with a value of $q$ that is close to the unconstrained maximum. Detailed results (on knapsack type problems) are presented in the supplement.

Figures 2–4 show the results for easy problems. For each network, the first two plots (in sub-figures (a) and (b)) show the impact of increasing time for two randomly chosen $q$ values and the last plot (in sub-figure (c)) shows the impact of increasing $q$ for a given time bound. The results are averaged over 10 runs. We observe that smaller values of $k$, specifically $k = 3, 5$ perform the best overall. The performance of $k = 1$ is only slightly inferior to $k = 3, 5$ on average while the performance of $k = 7, 9$ is 1-2 orders of magnitude inferior to $k = 3, 5$. We also observe that the performance of higher values of $k$ improves with time while the performance of smaller values of $k$ does not significantly improve with time. SCIP is substantially worse than our proposed algorithms.

Figures 5–7 show the results for hard problems for three different (network, $q$) pairs. To avoid clutter, we have excluded SCIP (since its performance is much worse than our algorithm). We observe that higher values of $k$, specifically $k = 7, 9$ perform the best overall. $k = 1$ is the worst performing scheme. As before, we see that the performance of higher values of $k$ improves with time while the performance of smaller values of $k$ does not significantly improve with time.

The discrepancy between the results for easy and hard problems can be "explained away" using the following intuitive arguments. As $k$ increases the number of nodes explored goes down which negatively impacts the performance (because the complexity of constructing the MCKP sub-problem is exponential in $k$). However, assuming that the MCKP solution obtained using greedy MCKP algorithms is close to optimal, as $k$ increases, we have access to a high quality solution to an exponentially increasing sub-problem. This positively impacts the performance. In other words, $k$

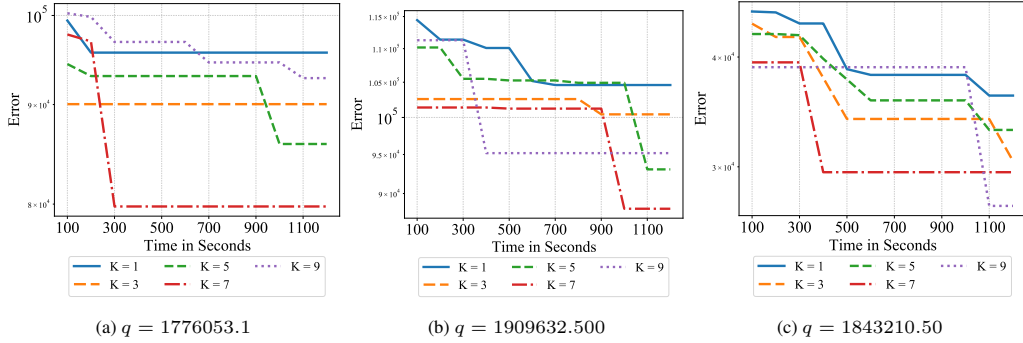

(a) $q = 1776053.1$     (b) $q = 1909632.500$     (c) $q = 1843210.50$

Figure 6: Hard problems: Results on (a) Grids_12 Markov network with 100 variables and 200 potentials, (b) Grids_13 Markov network with 100 variables and 300 potentials, and (c) Grids_14 Markov network with 100 variables and 300 potentials.

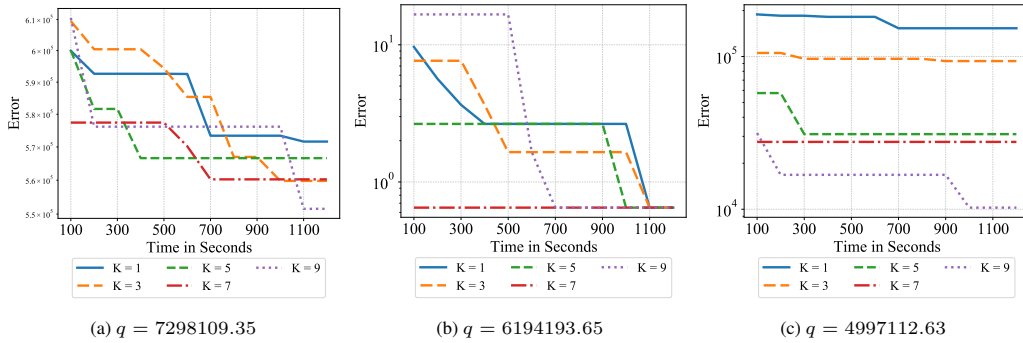

(a) $q = 7298109.35$     (b) $q = 6194193.65$     (c) $q = 4997112.63$

Figure 7: Hard problems: Results on (a) Grids_16 Markov network with 400 variables and 1160 potentials, (b) Grids_18 Markov network with 400 variables and 1160 potentials, and (c) Segmentation_15 Markov network with 232 variables and 853 potentials.

helps us explore the classic "exploration versus exploitation" trade off. When $k$ is small, the algorithm focuses on exploration while when $k$ is large, the algorithm spends more time on each state, exploiting good performing schemes having high computational complexity. Exploration is more beneficial on easy problems since there are many close to optimal solutions. On the other hand, for hard problems, exploitation is more beneficial because there are very few close to optimal solutions and it is easy to miss them or spend exponential time exploring them.

## 6 Conclusion and Future Work

In this paper, we presented a novel approach for solving the constrained most probable explanation (CMPE) problem in probabilistic graphical models. This problem is strongly NP-hard in general. We showed that developing advanced solvers for this problem is important because several explanation and estimation tasks can be reduced to it. The key idea in our approach is to condition on a subset of variables such that the remaining sub-problem can be encoded as a multiple choice knapsack (subset sum) problem, a weakly NP-hard problem that admits several efficient approximation algorithms. We showed that we can reason about the optimal subset of variables to condition on using a graph concept called $k$-separator. This allowed us to define powerful heuristic approximations to CMPE and analyze their computational complexity. Experiments on several benchmark networks showed that our algorithm is superior to SCIP, a state-of-the-art open source MILP solver. Our experiments also showed that when time is limited, higher values of $k$ are beneficial for hard CMPE problems while smaller values are beneficial for easy CMPE problems.

Future work includes: developing approximate dynamic programming algorithms; developing branch and bound algorithms by leveraging the mixed networks framework [25] and AND/OR search [8, 23, 24]; developing techniques for solving the constrained *marginal* most probable explanation problem; extending our approach to solve the same decision probability task [5], etc.

# 7 Broader Impact

Our work has mostly theoretical value and will require substantial effort and human-power in order to be used for commercial, government or defense purposes. The presented CMPE problem has potential applications in many sub-fields of AI and machine learning (ML) including computer vision, robotics, economics, operations research, NLP, and computational Biology where graphical models are used to represent and reason about structural features and uncertainty. The algorithm developed in this paper can be immediately leveraged to solve hard reasoning problems in these domains.

Apart from these obvious advantages that any optimization algorithm can bring to bear, as scientific research evolves, our work and technology can be misused deliberately or unknowingly. For example, finding the most likely assignment to all unobserved variables given evidence such that the model makes a specific decision is an application of our research. While it benefits the society via its superior prediction ability and potentially improving users' trust in the system, it can be misused by making a model make decisions in favor of a special group of people and harm the vulnerable ones (by appropriately modifying the constraints in CMPE), specifically in financial organizations. Our research can potentially lead to a practical tool which helps physicians diagnose diseases in a robust manner and fill up appropriate prescriptions by resolving conflicts. However, if the prior knowledge expressed in the graphical model is wrong or the relationships are learned inaccurately or the approximation error of our proposed algorithm is large (e.g., the problem belongs to one of the hard cases described in the experiments section), it may lead physicians to make a wrong decision/diagnosis. The negative consequences of this can be quite dire.

Today, a wide variety of tasks which are special cases of CMPE are solved by hand. In particular, several local experts who understand problem structure and who use implicit heuristic algorithms for solving special cases of CMPE are employed by various small businesses such as mom-and-pop moving companies, florists and local food delivery companies (not Grubhub). An advanced, easy to use tool for CMPE will obviate the need for local experts and may lead to significant job losses. Although the regulation and legal systems supported by governments along with developing profound knowledge about application fields can significantly manage potential harmful effects of such job losses (e.g., universal basic income), some damages are naturally inevitable.

## Acknowledgments

This work was supported in part by the DARPA Explainable Artificial Intelligence (XAI) Program under contract number N66001-17-2-4032, and by the National Science Foundation grants IIS-1652835 and IIS-1528037. Any opinions, findings, conclusions or recommendations expressed in this paper are those of the authors and do not necessarily reflect the views or official policies, either expressed or implied, of DARPA, NSF or the US government.

## Footnotes

[1]Note that this is not the same as computing the most probable assignment $\mathbf{H} = \mathbf{h}^*$ given $\mathbf{e}$ and $c$. For example, if $\sum_{f \in \mathbf{f}} f(\mathbf{h}^*, c, \mathbf{e}) < \sum_{f \in \mathbf{f}} f(\mathbf{h}^*, 1 - c, \mathbf{e})$ then $(\mathbf{h}^*, \mathbf{e})$ will flip the decision from $c$ to $1 - c$.

[2]Note that the number of vertices in the optimal $k$-separator of a graph can be quite large even if its treewidth is bounded by $k$. For instance, a complete binary tree has treewidth of 1 but the number of vertices in its 1-separator is bounded by $O(2^{h-1})$ where $h$ is the height of the tree.

[3]It is straight forward to encode CMPE as MILP (cf. [20]). We also experimented with Gurobi [17]. Its performance was inferior to SCIP because of precision problems.

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
