[Supplementary Material]

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

 problem of computing the most probable assignment for a decision can be reduced to CMPE.

## 3.1 Nearest Assignment/Explanation Problem

Given a graphical model $\mathcal{M} = \langle \mathbf{X}, \mathbf{f}, G \rangle$ and a real number $q$, the nearest assignment problem (NAP) is to find an assignment $\mathbf{x}$ to all variables in $\mathbf{X}$ such that $|q - \sum_{f \in \mathbf{f}} f(\mathbf{x})|$ is minimized. We can express NAP as CMPE using the following transformation. For each function $f \in \mathbf{f}$, let $g$ be a function defined as follows: $g(\mathbf{y}) = f(\mathbf{y}) - q/m$, where $\mathbf{y}$ is an assignment of values to all variables $\mathbf{Y} = S(f)$ and $m$ is the number of log-potentials. Let $\mathbf{x}^l$ and $\mathbf{x}^u$ be two assignments defined as follows:

$$\mathbf{x}^l = \max_{\mathbf{x}} \sum_{i=1}^{m} g_i(\mathbf{x}) \text{ s.t.} \sum_{i=1}^{m} g_i(\mathbf{x}) \leq 0 \quad \text{and} \quad \mathbf{x}^u = \max_{\mathbf{x}} \sum_{i=1}^{m} -g_i(\mathbf{x}) \text{ s.t.} \sum_{i=1}^{m} -g_i(\mathbf{x}) \leq 0$$

We can show that:

**Proposition 1.** $\min_{\mathbf{x}} |q - \sum_{f \in \mathbf{f}} f(\mathbf{x})|$ where $\mathbf{x} \in \{\mathbf{x}^l, \mathbf{x}^u\}$ *is the nearest assignment.*

*Proof.* Given a graphical model $\mathcal{M} = \langle \mathbf{X}, \mathbf{f}, G \rangle$ and a real number $q$, the nearest assignment problem (NAP) can be mathematically stated as

$$\min_{\mathbf{x}} |\sum_{i=1}^{m} f_i(\mathbf{x}) - q| \tag{3}$$

We can remove the absolute value requirement in the objective function using the following standard approach: convert the unconstrained minimization problem given above into a constrained minimization problem. In particular, we can define two nearest assignments $\mathbf{x}^l$ and $\mathbf{x}^u$ such that $\mathbf{x}^l$ maximizes $\sum_{f \in \mathbf{f}} f(\mathbf{x}) - q$ under the constraint that the value of the objective function is smaller than or equal to $0$ while $\mathbf{x}^l$ minimizes the same objective function under the constraint that the value of the objective function is larger than or equal to $0$. Formally, Eq. (3) can be rewritten as:

$$\min_{\mathbf{x}} |\sum_{i=1}^{m} f_i(\mathbf{x}) - q| = \min_{\mathbf{x} \in \{\mathbf{x}^l, \mathbf{x}^u\}} |\sum_{i=1}^{m} f_i(\mathbf{x}) - q| \tag{4}$$

where $\mathbf{x}^l$ and $\mathbf{x}^u$ are given by:

$$\mathbf{x}^l = \max_{\mathbf{x}} \left( \sum_{i=1}^{m} f_i(\mathbf{x}) - q \right) \text{ s.t. } \sum_{i=1}^{m} f_i(\mathbf{x}) - q \leq 0 \tag{5}$$

$$\mathbf{x}^u = \min_{\mathbf{x}} \left( \sum_{i=1}^{m} f_i(\mathbf{x}) - q \right) \text{ s.t. } \sum_{i=1}^{m} f_i(\mathbf{x}) - q \geq 0 \tag{6}$$

Given $g_i(\mathbf{y}) = f_i(\mathbf{y}) - q/m$, where $\mathbf{y}$ is an assignment of values to all variables $\mathbf{Y} = S(f_i)$ and replacing minimization as maximization (via negation), we can rewrite Eqs. (5) and (6) as:

$$\mathbf{x}^l = \max_{\mathbf{x}} \sum_{i=1}^{m} g_i(\mathbf{x}) \text{ s.t. } \sum_{i=1}^{m} g_i(\mathbf{x}) \leq 0 \tag{7}$$

$$\mathbf{x}^u = \max_{\mathbf{x}} \sum_{i=1}^{m} -g_i(\mathbf{x}) \text{ s.t. } \sum_{i=1}^{m} -g_i(\mathbf{x}) \leq 0 \tag{8}$$

Eqs. (4), (7) and (8) prove the proposition. $\qquad\square$

By inspection, the expressions for $\mathbf{x}^l$ and $\mathbf{x}^u$ are CMPE tasks. Thus NAP can be solved by solving two CMPE tasks. Rouhani et al.[31] describe an approximation algorithm that uses a 0-cutset [4] to solve NAP. Our work differs from the work of Rouhani et al. [31] in three ways:

- 0-cutsets are equivalent to 1-separators. Thus, our general-purpose algorithm can be seen as a generalization of Rouhani et al.'s approach to arbitrary $k$-separators. Our approach allows us to define more sophisticated approximation algorithms as a result of this generalization, including FPTAS algorithms for a more general class of graphical models (see section **??**).

- Rouhani et al.'s approach is not applicable to variables having non-binary domains while our proposed approach does not have such limitations.

- Rouhani et al.'s approach can be used if and only if the graphical model in the objective function is identical to the graphical model in the cost constraint. Our approach allows different graphical models to be present in the objective and cost constraint.

It turns out that NAP instances are one of the hardest CMPE problems. This is because NAP has subset-sum type constraints; it is well known that most pruning and bounding techniques (e.g., dominating items [20], linear programming relaxations) perform poorly on subset-sum problems.

## 3.2 Most Probable Assignment for a Decision

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

}_2$ with $c$ connected components where $\mathbf{X}_i$ is the set of variables in $i$-th connected component and $\mathbf{x}_i$ is an assignment to all variables of $\mathbf{X}_i$. Let $g_1, \ldots, g_c$ and $h_1, \ldots, h_c$ denote the functions obtained from $\mathcal{M}_1$ and $\mathcal{M}_2$ s.t. for $i = 1, \ldots, c$,

$$g_i(\mathbf{x}_i) = \sum_{f \in \mathbf{f}_1 : S(f) \subseteq \mathbf{X}_i} f(\mathbf{x}_i) \quad \text{and} \quad h_i(\mathbf{x}_i) = \sum_{t \in \mathbf{f}_2 : S(t) \subseteq \mathbf{X}_i} t(\mathbf{x}_i)$$

then, the CMPE problem can be mathematically stated as:

$$\max_{\mathbf{x}} \sum_{i=1}^{c} g_i(\mathbf{x}_i) \quad \text{s.t.} \quad \sum_{i=1}^{c} h_i(\mathbf{x}_i) \leq q' \tag{11}$$

Without loss of generality, let $M_i$ denote the number of entries in the functions $g_i$ and $h_i$. Let $Y_{ij}$ denote a Boolean variable which takes the value 1 if $\mathbf{X}_i = \mathbf{x}_i$ where $j \in \{1, \ldots, M_i\}$. Because, $\mathbf{X}_i$ has to be assigned exactly one value (and cannot take multiple values simultaneously), we have $\sum_{j=1}^{M_i} y_{ij} = 1$ where $y_{ij}$ is a 0/1 value assigned to $Y_i$. Let $s_{ij}$ and $r_{ij}$ denote the value of the $j$-th entry in $g_i$ and $h_i$ respectively. Then we can rewrite $g_i$ and $h_i$ as:

$$g_i(\mathbf{x}_i) = \sum_{j=1}^{M_i} y_{ij} s_{ij} \quad \text{s.t.} \quad \sum_{j=1}^{M_i} y_{ij} = 1 \tag{12}$$

$$h_i(\mathbf{x}_i) = \sum_{j=1}^{M_i} y_{ij} r_{ij} \quad \text{s.t.} \quad \sum_{j=1}^{M_i} y_{ij} = 1 \tag{13}$$

Substituting Eqs. (12) and (13) in Eq. (11), we can define the CMPE problem as follows:

$$\max_{\mathbf{y}} \sum_{i=1}^{c} \sum_{j=1}^{M_i} y_{ij} s_{ij} \quad \text{s.t.} \quad \sum_{i=1}^{c} \sum_{j=1}^{M_i} y_{ij} r_{ij} \leq q' \quad \text{and} \quad \sum_{j=1}^{M_i} y_{ij} = 1, \ i = 1, \ldots, c \tag{14}$$

Comparing the equation for CMPE given in Eq. (11) with the equation for MCKP given in Eq. (**??**), it is easy to see that the two problems are equivalent under the following substitutions: $c = m$ (the number of components equals the number of bins), $x_{ij} = y_{ij}$ (one-to-one mapping between the Boolean variables), $r_{ij} = p_{ij}$ (each entry in $g_i$ corresponds to the profit of an item), $s_{ij} = c_{ij}$ (each entry in $h_i$ corresponds to the cost of an item), and $q' = q$. Under this substitution, it is easy to see that a (feasible) solution to the CMPE problem can be recovered from a solution to the MCKP in $O(n)$ time where $n$ is the number of variables in $\mathcal{M}_1$ (and $\mathcal{M}_2$). $\qquad\square$

Since the number of items in each bin $i$ equals the number of entries in $g_i$, the number of items in bin $i$ is exponential in the number of variables in the scope of $g_i$, namely it equals $\exp(|\mathbf{X}_i|)$. Thus Encoding 4.2 will yield a bounded MCKP if $|\mathbf{X}_i|$ is bounded by a constant for all $i$.

**(a)**

$\mathcal{M}_1, \mathcal{M}_2$

(Combined primal graph with vertices $X_1, X_2, X_4, X_3, X_5$; edges $X_1$–$X_2$, $X_1$–$X_3$, $X_2$–$X_3$, $X_4$–$X_5$.)

**(b)**

| $X_1$ | $X_2$ | $X_3$ | $g_1$ | $h_1$ |
|---|---|---|---|---|
| 0 | 0 | 0 | 5 | 1 |
| 0 | 0 | 1 | 5 | 5 |
| 0 | 1 | 0 | 9 | 4 |
| 0 | 1 | 1 | 3 | 3 |
| 1 | 0 | 0 | 9 | 8 |
| 1 | 0 | 1 | 2 | 3 |
| 1 | 1 | 0 | 6 | 7 |
| 1 | 1 | 1 | 2 | 9 |

$g_1, h_1$

| $X_4$ | $X_5$ | $g_2$ | $h_2$ |
|---|---|---|---|
| 0 | 0 | 1 | 8 |
| 0 | 1 | 9 | 8 |
| 1 | 0 | 4 | 4 |
| 1 | 1 | 6 | 1 |

$g_2, h_2$

$q = 10$

**(c)**

| $j$ | $p_{1j}$ | $c_{1j}$ |
|---|---|---|
| 1 | 5 | 1 |
| 2 | 5 | 5 |
| 3 | 9 | 4 |
| 4 | 3 | 3 |
| 5 | 9 | 8 |
| 6 | 2 | 3 |
| 7 | 6 | 7 |
| 8 | 2 | 9 |

$bin_1$

| $j$ | $p_{2j}$ | $c_{2j}$ |
|---|---|---|
| 1 | 1 | 8 |
| 2 | 9 | 8 |
| 3 | 4 | 4 |
| 4 | 6 | 1 |

$bin_2$

$capacity = 10$

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

We summarize the discussion above in the following proposition and two corollaries.

**Proposition 3.** *If there exists an algorithm for solving MCKP with approximation factor $0 \le \alpha \le 1$ having time complexity $O(t)$ then Algorithm* ANYTIME-CMPE *has approximation factor $\alpha$ and has time complexity $O((n + c \exp(k) + t) \times \exp(|\mathbf{S}|))$ under the assumption that systematic enumeration is used to search over value assignments to $\mathbf{S}$ (and the algorithm is terminated after all assignments to $\mathbf{S}$ are enumerated).*

*Proof.* Without loss of generality, let $\{\mathbf{s}^1, \ldots, \mathbf{s}^a\}$ denote the set of feasible solutions of CMPE projected over the set $\mathbf{S}$. For each feasible solution $\mathbf{s}^i$, let

$$f^i = \sum_{f \in \mathbf{f}_1 : S(f) \subseteq \mathbf{S}} f(\mathbf{s}^i)$$

and

$$g^i = \max_{\mathbf{y}} \sum_{j=1}^{c} \sum_{f \in \mathbf{f_1} : S(f) \cap \mathbf{X}_j \neq \emptyset} f(\mathbf{y}, \mathbf{s}^i)$$

Then, the optimal value of the objective function for the CMPE problem is given by

$$\text{OPT} = \max_{1 \leq i \leq a} \left\{ f^i + g^i \right\} \tag{15}$$

Given an algorithm A for MCKP having approximation factor $0 \leq \alpha \leq 1$, let $s^i \geq \alpha g^i$ denote the solution output by A. We have:

$$\max_{1 \leq i \leq a} \left\{ f^i + s^i \right\} \geq \max_{1 \leq i \leq a} \left\{ f^i + \alpha g^i \right\} \geq \alpha \max_{1 \leq i \leq a} \left\{ f^i + g^i \right\} = \alpha \text{OPT}$$

Since the time complexity of the given algorithm is $O(t)$ and the time complexity of constructing the MCKP and retrieving its solution is $O(n + c \exp(k))$, the overall time complexity of lines 6–14 given an assignment to $\mathbf{S}$ is $O(n + c \exp(k) + t)$. Thus, the overall time complexity is $O((n + c \exp(k) + t) \times \exp(|\mathbf{S}|))$. □

We can derive the following two corollaries from Proposition 3.

**Corollary 1.** *Given a constant $k \geq 1$, there exists a polynomial time algorithm (polynomial in $n$) having an approximation factor $4/5$ for solving the CMPE problem under the assumption that the size of the $k$-separator for the combined primal graph of $\mathcal{M}_1$ and $\mathcal{M}_2$ is bounded by a constant.*

*Proof.* Proof follows from Gens and Levner [15] and Proposition 3. □

**Corollary 2.** *Given a constant $k \geq 1$ and $0 \leq \epsilon \leq 1$, there exists a FPTAS which produces a $(1 - \epsilon)$ approximate solution and runs in polynomial time in $n$ and $1/\epsilon$ for solving the CMPE problem under the assumption that the size of the $k$-separator for the combined primal graph of $\mathcal{M}_1$ and $\mathcal{M}_2$ is bounded by a constant.*

*Proof.* Proof follows from Bansal et al. [1] and Proposition 3. □

## 5   Experiments

### 5.1   Setup

We compared the performance of Algorithm ANYTIME-CMPE with SCIP [14], a state-of-the-art open source mixed integer linear programming (MILP) solver.[3] We evaluated the impact of increasing $k$ and time on the performance of ANYTIME-CMPE. We experimented with the following five values of $k$: $\{1, 3, 5, 7, 9\}$. For each $k$, we ran our algorithm on each probabilistic network for 1200 seconds. SCIP was also run for 1200 seconds on each network.

**Implementation Details.** We used restart-based local search to perform search over the value assignments to $\mathbf{S}$. Specifically, our implementation makes locally optimal moves if it improves the evaluation score. Otherwise, the local search is stuck in local maxima and we make a random move. We implemented the greedy MCKP algorithm of [15] to solve $P_{\mathbf{s}}$. We improve the greedy solution further by performing local search over $P_{\mathbf{s}}$ until a local maxima is reached. We used the max-degree heuristic outlined in section 4.2 to select a minimal $k$-separator.

**Benchmarks and Methodology.** We experimented with the following benchmark graphical models, available from the UAI 2010 and 2014 competitions [12, 17]: (1) Large Dynamic Bayesian Networks, (2) Ising models and (3) Image Segmentation networks. For each benchmark network, we selected ten $q$ values as follows. We generated a million assignments uniformly at random and divided their weights into 10 quantiles (deciles). Then, we selected a random weight from each of the 10 quantiles as a value for $q$. We found that most CMPE problems generated this way were easy problems in that the maximum value of the objective function (or close to it) was reached quickly by all of our local search algorithms. Similar observations have been made in the literature on knapsack problems

(a) $q = -27.0269$                 (b) $q = -0.008$                 (c) Time = 1200 seconds

Figure 2: Easy problems: Results on DBN_16 Markov network having 44 variables and 528 potentials.

(a) $q = -34.287$                 (b) $q = 0.015$                 (c) Time = 1200 seconds

Figure 3: Easy problems: Results on Grids_11 Markov network having 100 variables and 300 potentials.

(a) $q = -365.795$                 (b) $q = -19.484$                 (c) Time = 1200 seconds

Figure 4: Easy problems: Results on Grids_12 Markov network having 100 variables and 280 potentials.

[29]. Therefore, in order to generate hard CMPE problems, we made the following modifications: (1) for each network, we kept the network structure the same but changed the parameters by sampling each parameter from the range $[0, 10000]$; (2) we selected values of $q$ that are close to the weight of the (unconstrained) most probable assignment; and (3) we focused on multiple choice subset sum problems, namely we chose $\mathcal{M}_1 = \mathcal{M}_2$. We use the quantity $q - o$, which we call error to measure performance when $\mathcal{M}_1 = \mathcal{M}_2$ where $o$ is value of the objective function output by the competing algorithms. Note that the maximum value of the objective function is bounded by $q$ (since we are solving hard subset sum type problems) and therefore smaller the error, better the algorithm. We used the value of the objective function $o$ as our performance measure for knapsack-type problems (since $\mathcal{M}_1 \neq \mathcal{M}_2$, the value of the objective function is not upper bounded by $q$). Thus for knapsack-type problems, higher the value of the performance measure, better the algorithm.

## 5.2   Results on Subset-Sum Type Problems ($\mathcal{M}_1 = \mathcal{M}_2$)

We present results for two classes of problems: (1) relatively *easy* subset-sum type problems on the original networks; and (2) *hard* subset-sum type problems on the modified networks (only the

(a) $q = -559.495$  (b) $q = 0564.392$  (c) Time = 1200 seconds

Figure 5: Easy problems: Results on Grids_14 Markov network having 100 variables and 300 potentials.

(a) $q = -642.153$  (b) $q = -531.726$  (c) Time = 1200 seconds

Figure 6: Easy problems: Results on Segmentation_12 Markov network with 229 variables and 851 potentials.

(a) $q = -561.619$  (b) $q = -551.510$  (c) Time = 1200 seconds

Figure 7: Easy problems: Results on Segmentation_14 Markov network with 226 variables and 845 potentials.

(a) $q = 6684822.85$  (b) $q = 4845547.43$  (c) $q = 4733941.8$

Figure 8: Hard problems: Results on (a) Grids_17 Markov network with 400 variables and 1160 potentials, (b) Segmentation_12 Markov network with 229 variables and 851 potentials, and (c) Segmentation_14 Markov network with 226 variables and 845 potentials.

Figure 9: Hard problems: Results on (a) Grids_12 Markov network with 100 variables and 200 potentials, (b) Grids_13 Markov network with 100 variables and 300 potentials, and (c) Grids_14 Markov network with 100 variables and 300 potentials.

Figure 10: Hard problems: Results on (a) Grids_16 Markov network with 400 variables and 1160 potentials, (b) Grids_18 Markov network with 400 variables and 1160 potentials, and (c) Segmentation_15 Markov network with 232 variables and 853 potentials.

parameters are modified as described above) with a value of $q$ that is close to the unconstrained maximum.

Fig. 2-7 show the results for easy problems. For each network, the first two plots (in sub-figures (a) and (b)) show the impact of increasing time for two randomly chosen $q$ values and the last plot (in sub-figure (c)) shows the impact of increasing $q$ for a given time bound. The results are averaged over 10 runs. We observe that smaller values of $k$, specifically $k = 3, 5$ perform the best overall. The performance of $k = 1$ is only slightly inferior to $k = 3, 5$ on average while the performance of $k = 7, 9$ is 1-2 orders of magnitude inferior to $k = 3, 5$. We also observe that the performance of higher values of $k$ improves with time while the performance of smaller values of $k$ does not significantly improve with time. SCIP is substantially worse than our proposed algorithms.

Fig. 8-10 show the results for hard problems for different (network, $q$) pairs. To avoid clutter, we have excluded SCIP (since its performance is much worse than our algorithm). We observe that higher values of $k$, specifically $k = 7, 9$ perform the best overall. $k = 1$ is the worst performing scheme. As before, we see that the performance of higher values of $k$ improves with time while the performance of smaller values of $k$ does not significantly improve with time.

The discrepancy between the results for easy and hard problems can be "explained away" using the following intuitive arguments. As $k$ increases the number of nodes explored goes down which negatively impacts the performance (because the complexity of constructing the MCKP sub-problem is exponential in $k$). However, assuming that the MCKP solution obtained using greedy MCKP algorithms is close to optimal, as $k$ increases, we have access to a high quality solution to an exponentially increasing sub-problem. This positively impacts the performance. In other words, $k$ helps us explore the classic "exploration versus exploitation" trade off. When $k$ is small, the algorithm focuses on exploration while when $k$ is large, the algorithm spends more time on each state, exploiting good performing schemes having high computational complexity. Exploration is more beneficial on easy problems since there are many close to optimal solutions. On the other hand, for hard problems,

Figure 11: Knapsack-type problems: Results on (a) DBN_11 Markov network with 40 variables and 440 potentials, (b) DBN_15 Markov network with 42 variables and 483 potentials, and (c) DBN_16 Markov network with 44 variables and 528 potentials.

Figure 12: Knapsack-type problems: Results on DBN_12 Markov network with 42 variables and 483 potentials.

Figure 13: Knapsack-type problems: (a), (b): Results on DBN_13 Markov network with 44 variables and 528 potentials, and (c) Results on Grids_11 Markov network with 100 variables and 300 potentials.

exploitation is more beneficial because there are very few close to optimal solutions and it is easy to miss them or spend exponential time exploring them.

## 5.3  Results on Knapsack Type Problems ( $\mathcal{M}_1 \neq \mathcal{M}_2$ )

To demonstrate the performance of Algorithm ANYTIME-CMPE on MCKP-type problems, (where $\mathcal{M}_1 \neq \mathcal{M}_2$ ) we designed and performed another set of experiments. To generate MCKP-type problems, we made a copy of each benchmark network and randomly perturbed each parameter by $\pm\epsilon$ where $\epsilon$ is sampled uniformly at random from $(0, 1]$ ; we used the same network structure as the original one. We used the same approach as the one used in subset-sum problems for generating values of $q$. As mentioned earlier, we used the value of the objective function output by the competing algorithms as our performance measure; higher the value better the algorithm.

We used the same setup as subset-sum problems: $k \in \{1, 3, 5, 7, 9\}$ and for each $k$, we ran our algorithm on each pair of network and $q$ for 1200 seconds. Also, we used SCIP as a baseline and

compared the output of ANYTIME-CMPE with it. We evaluated the impact of increasing $k$ value and the running time on the performance of our algorithm.

Fig. 11-13 show the results of our experiments for knapsack-type problems. Each figure plots the value of the objective function as a function of time and shows the impact of increasing $k$ and time for a given $q$ values. Each figure also shows the impact of increasing time on the performance of SCIP. The plots are averaged over 10 runs.

Similar to easy subset-sum type problems, we observe that the difference between the values of objective function computed by various algorithms is small (notice the range on the Y-axis of the figures, it varies between $0.1 - 0.3$ for most plots). Our experiments show that in problems with relatively large objective values, different $k$ values perform the same and SCIP is far worse than our algorithm. On the other hand, in problems with relatively small objective values, where potential values are more close to each other, large values of $k$ (e.g., $k = 7$) are superior to SCIP and small values of $k$.

## 6 Conclusion and Future Work

In this paper, we presented a novel approach for solving the constrained most probable explanation (CMPE) problem in probabilistic graphical models. This problem is strongly NP-hard in general. We showed that developing advanced solvers for this problem is important because several explanation and estimation tasks can be reduced to it. The key idea in our approach is to condition on a subset of variables such that the remaining sub-problem can be encoded as a multiple choice knapsack (subset sum) problem, a weakly NP-hard problem that admits several efficient approximation algorithms. We showed that we can reason about the optimal subset of variables to condition on using a graph concept called $k$-separator. This allowed us to define powerful heuristic approximations to CMPE and analyze their computational complexity. Experiments on several benchmark networks showed that our algorithm is superior to SCIP, a state-of-the-art open source MILP solver. Our experiments also showed that when time is limited, higher values of $k$ are beneficial for hard CMPE problems while smaller values are beneficial for easy CMPE problems.

Future work includes: developing approximate dynamic programming algorithms; developing branch and bound algorithms by leveraging the mixed networks framework [26] and AND/OR search [9, 24, 25]; developing techniques for solving the constrained *marginal* most probable explanation problem; extending our approach to solve the same decision probability task [6], etc.

## Acknowledgments

This work was supported in part by the DARPA Explainable Artificial Intelligence (XAI) Program under contract number N66001-17-2-4032, and by the National Science Foundation grants IIS-1652835 and IIS-1528037. Any opinions, findings, conclusions or recommendations expressed in this paper are those of the authors and do not necessarily reflect the views or official policies, either expressed or implied, of DARPA, NSF or the US government.

## Footnotes

[1]Note that this is not the same as computing the most probable assignment $\mathbf{H} = \mathbf{h}^*$ given $\mathbf{e}$ and $c$. For example, if $\sum_{f \in \mathbf{f}} f(\mathbf{h}^*, c, \mathbf{e}) < \sum_{f \in \mathbf{f}} f(\mathbf{h}^*, 1 - c, \mathbf{e})$ then $(\mathbf{h}^*, \mathbf{e})$ will flip the decision from $c$ to $1 - c$. In terms of complexity, computing the most probable explanation (MPE) given evidence can be solved in polynomial time on bounded treewidth networks. CMPE, on the other hand, is NP-hard on bounded treewidth networks (reduction from MCKP).

[2]Note that the number of vertices in the optimal $k$-separator of a graph can be quite large even if its treewidth is bounded by $k$. For instance, a complete binary tree has treewidth of 1 but the number of vertices in its 1-separator is bounded by $O(2^{h-1})$ where $h$ is the height of the tree.

[3]It is straight forward to encode CMPE as MILP (cf. [21]). We also experimented with Gurobi [18]. Its performance was inferior to SCIP because of precision problems.