[Reviews · NeurIPS 2020]

Review 1

Summary and Contributions: The authors propose a new approximate optimization method for the constrained max. a posteriori inference problem in discrete graphical models. The single constraint has a form f(x_1, ... , x_n) <= c, where x_1,..., x_n are the discrete variables of the considered graphical model. The proposed method consists in - considering a combined graph, which includes all edges from the original graphical model and the edges corresponding to the function f, if one considers it as an energy of a different graphical model; - fixing the values of a subset of variables such that the remaining graph falls into multiple small connected components, up to a bounded k number of variables each. - a reduction of the problem on the remaining graph to the multiple choice knapsack problem, where each connected component is treated as a single variable. - for solving the multiple choice knapsack problem an existing algorithm is used.

Strengths: The paper is theoretically sound and mathematical statements are rigorously proven. The problem is - important, as a number of existing problems can be reduced to it - NP-hard, as the max. a posteriori inference is NP-hard by itself Therefore, any progress in this direction is important. The proposed reduction is new for me. The discrete graphical models is one of the important topics for the NeurIPS community.

Weaknesses: The proposed reduction is practically limited to the case when the initial graph and the graph defined by the function f are sparse and the number of variables, which must be fixed to obtain small connected components, is relatively small. As a consequence, the empirical evaluation is limited to the small-sized sparse problems from UAI 2010 and 2014 competitions.

Correctness: The claims and the method are correct. The empirical evaluation is sensible, I was unable to propose a different evaluation methodology on the fly.

Clarity: The paper is written very clearly and rigirously.

Relation to Prior Work: The relation to the previous work is clearly stated.

Reproducibility: Yes

Additional Feedback: Major: - Why the SCIP solver was used as a baseline and not Gurobi? My experience suggests that Gurobi typically performs notably better and the academic license is free. - Proposition 1 does not contain a claim. It seems you unintentionally deleted a part of it. Minor: - I would suggest to use the term "volume" instead of "cost", as it makes more sense to restrict the volume and maximize the profit than to restrict the costs and maximize the profit, especially when one speaks about a knapsack (with a predefined volume). l117: m is the number of log-potentials. It becomes more readable, if you would write "m is the number of log-potentials, i.e. m=|\mathbf(f)|". x^l = argmax ... , x^u = argmin ... l155: the sentence "and a real number q such that no two functions ... share any variable..." - is ambiguous. Put "and a real number q" right before "we can construct" instead. Section 4.3. Please specify the meaning of "n" again to simplify understanding. It is the number of connected components, is not it? =============Post rebuttal:================================ I've read the other reviews and the rebuttal of the authors. It seems to me that authors properly addressed the main concerns of all reviewers. Therefore, I leave my positive evaluation of the paper unchanged. As an additional remark to Proposition 1: 'argmax' instead of 'max' at the very beginning of the statement would eliminate my confusion.


Review 2

Summary and Contributions: The paper considers a class of discrete optimisation problems which requires to maximise the (negative) energy of a discrete probabilistic graphical model under the constraint that the energy of the solution w.r.t. a second model is bounded by some given value. The authors propose an algorithm that reduces this task to a set of multiple choice knapsack problems which are then solved by a greedy algorithm.

Strengths: The authors discuss several application relevant problem classes that can be reduced to the studied problem. The main idea of the proposed approach is quite unexpected and consists in following. The first step is to find a separator set in the combined graph of the two models, such that the remaining graph decomposes into connected components with at most k elements. After fixing a variable assignment for the separator set, the initial task can be reduced to a multiple choice knapsack problem, where the choices in the bins are the variable assignments of the connected components. The authors show that their approach outperforms of the shelf mixed ILP solvers considerably for several benchmark graphical models.

Weaknesses: The paper is clearly structured and concise. I see almost no weakness. Of course, all the sub-problems, i.e. (1) finding an optimal separator set, (2) enumerating over all possible label assignments on the separator set and (3) solving the multiple choice knapsack problem are themselves hard and require heuristic approaches. However, the paper shows that using well studied approximations for each of them leads, altogether, to an algorithm that outperforms standard solvers by orders of magnitude.

Correctness: The claims and proposed methods are correct. The experimental validation is convincing.

Clarity: The paper is clearly structured and well written.

Relation to Prior Work: Relation to prior work is clearly discussed. The proposed method is unexpected and novel.

Reproducibility: Yes

Additional Feedback: Post rebuttal: I have read the other reviews and the rebuttal of the authors. It seems to me that they have addressed the concerns raised in the other reviews. I will therefore keep my positive recommendation for accepting the paper.


Review 3

Summary and Contributions: The authors define and study a combinatorial optimization task called the "constrained most probable explanation" (CMPE) in discrete PGMs, to find the MPE configuration x of some distribution p(x) given a bound on another, q(x). The authors' approach is essentially to find a cutset of the graph of p & q such that the resulting independent components are small enough to be solved (approximately) using methods from multiple choice knapsack problems.

Strengths: The most interesting aspect of the work is the problem formulation itself, and its interpretation as a multiple choice knapsack problem. This was new to me, and seems like it would be of interest generally. The authors argue that the CMPE problem is important, since several tasks (m-best, nearest assignment, and decision-preserving MAP) can be framed as CMPE.

Weaknesses: The significance of the work is unclear, in my opinion. The authors do not study any realistic problems corresponding to CMPE tasks, or use the tasks that can be framed as CMPE to compare their approach with other methods for solving existing problems. The experiments look at the performance of synthetic CMPE problems defined on UAI benchmark instances, some of which are built from real data, but not for the purpose of a CMPE task. It would be much more compelling to see this applied to something more realistic. In terms of methodology, the approach is mainly a combination of two existing techniques (cutsets and MCKP solvers) and is thus mostly innovative in their application to the new task rather than as a novel approach.

Correctness: Yes. Empirical methodology appears correct, with the criticism that it is very synthetic in the manner noted.

Clarity: Yes, the paper is clear and well written.

Relation to Prior Work: Yes; the work is defining a new problem type.

Reproducibility: Yes

Additional Feedback: I think this work would be greatly strengthened by showing that these types of queries can be useful for something practical, and evaluating on real models, or models meant to simulate that type of task.


Review 4

Summary and Contributions: The paper presents a new constrained MAP inference task for graphical models. Specifically, given two graphical models defined over the same set of variables and a real number q, the task is to find a MAP assignment for the first graphical model such that the cost of the assignment is less than q with respect to the second graphical model. The proposed algorithm combines (local/exhaustive) search over a subset of variables while encoding the remaining conditioned subproblem as a multiple-choice knapsack problem and solving that with a specialised solver. The empirical evaluation on standard benchmarks for graphical models shows that the proposed algorithm performs well in practice and outperforms a MIP encoding of the problem that is subsequently solved with an open-source MIP solver.

Strengths: The constrained inference task defined in this paper appears to be relevant to other domains as well. Therefore, I think the paper presents a fairly strong contribution. Furthermore, the empirical evaluation is performed on standard benchmarks for graphical models and therefore I think the results can be easily reproduced.

Weaknesses: My main concern is that the paper is not really self contained. The proposed anytime algorithm relies on state-of-the-art MCKP solvers. So I think the background section should contain a brief description of such solvers.

Correctness: The technical contribution appears to be correct. I didn't see any obvious mistakes.

Clarity: The quality of the presentation is overall quite good. The technical details are discuss in a fairly clear manner and therefore the paper is relatively easy to follow.

Relation to Prior Work: The paper appears to be positioned well in the context of related work. However, I think there is a close connection with Dechter's "mixed networks" framework that also defines a constrained inference task over graphical models. Robert Mateescu and Rina Dechter. "Mixed deterministic and probabilistic networks". In Annals of Mathematics and Artificial Intelligence. Special Issue: Probabilistic Relational Learning, Volume 54 (1-3), pages 3-51, 2008.

Reproducibility: Yes

Additional Feedback: 1. I think it's important to present in the empirical section the MIP encoding of the proposed inference task. 2. I was surprised that no established MAP inference algorithms for graphical models have been pursued in the paper. For example, a straightforward approach is to run a depth-first branch and bound search algorithm for MAP that enumerates the MAP assignments of M1 and evaluates them on M2 to determine if they exceed the q cost or not. 3. It is well known that solving MAP as an integer program is not the most efficient way, so I'm not that surprised that SCIP doesn't perform that well. *** Post rebuttal *** I've read the other reviews and the author response. I'm satisfied with the clarifications provided by the authors.

[Author Response · NeurIPS 2020]

We thank all reviewers for their comments and appreciate the fact that all of you carefully read the paper (this is evident from your comments). We will address Reviewers 1, 2, 3 and 4 as **R1**, **R2**, **R3** and **R4** respectively.

**1. Sparsity.** (We say that a graph is sparse for CMPE if it has a bounded (small) $k$-separator for a given $k$). We will make two points in order to alleviate **R1**'s concern that "the proposed method is practically limited because it requires sparse graphs." First, notice that even though the UAI 2010 and 2014 instances have relatively small treewidth (roughly 15-50), their $k$-separator size can be quite large. In other words, the UAI instances are not sparse from the point of view of the CMPE problem and despite this our proposed method works relatively well (primarily because of the MCKP formulation). We will report the $k$-separator sizes in the paper as well as supplement and add a proposition about the relationship between $k$-separator and treewidth in the paper (**R1** and **R2**). Second (in future work), one can develop structure learning algorithms that induce graphical models having small $k$-separators from data, namely use the size of the $k$-separator as inductive bias (inspired by work in the tractable probabilistic models community where an upper bound on complexity of posterior marginal inference is used as inductive bias).

**2. Gurobi and MILP Encoding.** (**R1** and **R4**) We did not use Gurobi because of precision problems. We tried to play with the tolerances provided on the Gurobi website as well as scaling (see Gurobi manual). However, we found that the solutions Gurobi returned were often inferior to SCIP because tolerances/gaps in SCIP can be set to a much smaller value. To alleviate **R4**'s concerns, we will describe our MILP encoding in the extended version of the paper. C++ code for the encoding is already included in the supplementary material.

**3. Significance of CMPE and Experimental Results.** We agree with **R3**'s assessment that a more compelling case can be made with experiments on a concrete real world application. This is part of our future work. However, we believe that we have performed a systematic experimental study on *realistic sized probabilistic models* (used in past UAI competitions) as well as *hard problems* for our proposed method. All reviewers have rightly pointed out the significance of CMPE as a *unifying* query type because many reasoning queries in graphical models can be reduced to it.

**4. $k$-separators.** As far as we know, this is the first paper that uses the concept of $k$-separator for efficient inference in graphical models. It is related to a previously proposed concept called $w$-cutset, but not the same as the latter (**R3**).

**5. Background on MCKP.** (**R4**) An excellent reference for MCKP is the book on Knapsack problems by Kellerer et al. [1]. We have included multiple references for knapsack solvers in the paper (including the book above). However, in order to make the paper self contained, we will try to describe the specific solver used in more detail. Thank you for the suggestion. The good news is that (and as mentioned in the paper) we can leverage advances in MCKP solvers to improve the efficiency and scalability of CMPE solvers because of our proposed method.

**6. Relationship to Mixed Networks.** Notice that there is just one global constraint in CMPE. Therefore we did not use the mixed networks framework. The latter is useful when you have a number of local constraints defined over a subset of variables and efficient constraint propagation techniques (e.g., arc consistency, path consistency, etc.) exist for handling the local constraints. Moreover, in presence of local constraints, the problem can be solved (exactly) in time and space that scales exponentially with the treewidth of the combined primal graph. CMPE is a much harder task and remains NP-hard even on bounded treewidth combined primal graphs. (**R4**)

**7. Using MPE solvers for CMPE.** This approach will be very inefficient. The constraint in CMPE will cause minimal pruning and the search procedure will enumerate a large number of assignments. It will be only useful for assignments that are at or very near the (unconstrained) MPE value. Again, we want to emphasize that CMPE is much harder than MPE, both for bounding and solving. For example, the MPE solver approach will be inefficient even if the combined primal graph is empty. On empty graphs MPE can be solved in linear time while CMPE remains NP-hard. (**R4**)

**8. Other Minor Points Raised by the Reviewers.**

- Proposition 1 is a claim. It says that if you find two assignments $\mathbf{x}^u$ and $\mathbf{x}^l$ by solving the CMPEs defined in the equation above the proposition, then the nearest assignment is either $\mathbf{x}^u$ or $\mathbf{x}^l$. Thus, NAP can be solved if you solve CMPE. (**R1**)

- We agree that using "volume" instead of "cost" makes more sense. However, the term "cost" is often used in the Knapsack literature and we were just being consistent. (**R1**)

# References

[1] H. Kellerer, U. Pferschy, and D. Pisinger. *Knapsack Problems*. Springer, Berlin, Germany, 2004.


[Meta-Review · NeurIPS 2020]

The paper proposes a new inference task for graphical models. It consists in finding a MAP assignment w.r.t. one distribution p such that its probability w.r.t. another distribution q is bounded. It contains as special cases several interesting graphical model problems like m-best assigments. The method uses a transformation to multiple choice knapsack for cmputationally solving the problem. Authors agree that the new problem is interesting and the transformation to multiple choice knapsack is interesting. The main criticism pertains to the small experiments that are not necessarily indicative of real-world problems. The interesting new problem formulation and innovative solution technique still merit publication at NeurIPS.